# Molecular Characterization and Phylogenetic Analysis of Flightin Gene in *Vespa basalis* (Hymenoptera, Vespidae)

**DOI:** 10.3390/ani14060978

**Published:** 2024-03-21

**Authors:** Hasin Ullah, Xiaojuan Huang, Tong Zhou, Yan Tang, Danyang Zhu, Hongli Xu, Jiangli Tan

**Affiliations:** Shaanxi Key Laboratory for Animal Conservation/Key Laboratory of Resource Biology and Biotechnology in Western China, Ministry of Education, College of Life Sciences, Northwest University, Xi’an 710069, China; hasenullah888@yahoo.com (H.U.); huangxiaojuan@stumail.nwu.edu.cn (X.H.); zhoutong@stumail.nwu.edu.cn (T.Z.); tangyan1@stumail.nwu.edu.cn (Y.T.); zhudanyang@stumail.nwu.edu.cn (D.Z.); 202233084@stumail.nwu.edu.cn (H.X.)

**Keywords:** *Vespa basalis*, flightin, phylogenetic relationship, Vespidae, evolution

## Abstract

**Simple Summary:**

We conducted a study on flightin, a vital protein found in insect flight muscles, with a specific focus on the social wasp *Vespa basalis*, which is a dangerous hornet species. Using sequence analysis techniques, we successfully obtained an 1189 bp cDNA sequence encoding the flightin protein consisting of 150 amino acids. According to our analysis, the molecular weight and isoelectric point of the amino acid sequence are 18.05 kDa and 5.84, respectively. The protein lacks a transmembrane topology structure, and we identified four patterns of functional sites. Conducting phylogenetic analysis on 38 species, including 8 Vespidae species, revealed a close evolutionary relationship between *Vespa basalis* and *Vespa mandarinia*. These findings warrant further investigation, especially considering the contrasting information obtained from the analysis of mitochondrial sequences.

**Abstract:**

Flight is a complex physiological process requiring precise coordination of muscular contraction. A key protein in insect flight is flightin, which plays an integral role in the flight muscles. This research sought to evaluate the flight competence of the social wasp *V. basalis* by characterizing the molecular components involved. Our study focused on *Vespa basalis*, one of the most dangerous hornet species, utilizing PCR to obtain a partial cDNA sequence of the flightin protein. We then employed phylogenetic and sequence analysis to gain insights into this protein in flight-related adaptations. The cDNA has an 1189-base pair sequence including an open reading frame (453 bp) encoding 150 amino acids. Analyzing the deduced amino acid sequence using an online tool revealed a molecular weight of 18.05 kDa, an isoelectric point of 5.84, four functional site patterns, and no transmembrane topology. We constructed a phylogenetic tree of flightin based on 38 species. Our analysis indicated that *V. basalis* is most closely related to *V. mandarinia*; this alignment is consistent with their similar aggressive behavior, but their evolutionary relationship, based on mitochondrial sequences, presents a contrast. These initial findings on the flightin gene in *V. basalis* lay the groundwork for future functional studies to elucidate its specific role in flight adaptations and explore its potential as a target for pest management strategies.

## 1. Introduction

Social wasps within the subfamily Vespinae (Hymenoptera: Vespidae), including hornets and yellow jackets, play diverse ecological roles. They serve as essential predators, contributing to the preservation of natural balance in ecosystems and the control of agricultural and forestry pests [1]. However, their highly developed social colonies can pose a threat to humans, leading to potential attacks [2,3]. Among these species, *V. basalis* is one of the most prevalent, with a wide distribution in China and Southeast Asia. It is also known as one of the most aggressive hornets that pose a significant danger to animals [4].

Flight muscles are unique to insects, enabling them to navigate various environments [5,6]. The structure and development of insect flight muscles are crucial for their robust flight capabilities, as they convert chemical energy into mechanical energy [7,8,9,10]. These muscles are typically classified as direct flight muscles (DFM) or indirect flight muscles (IFM), depending on the type of wing motion they generate [11,12]. IFM contains myofibrillar proteins that are vital for flight and have been extensively studied in *Drosophila* [13]. Moreover, these proteins exhibit a high degree of conservation throughout insect evolution [14,15]. They play essential roles in constructing the thick filament, thin filament, Z disc, and connecting filament [16,17].

The protein flightin plays a crucial role in the functioning of IFM [13]. Studies conducted using *D. melanogaster* as a model organism have unveiled flightin’s responsibility for maintaining the structural integrity of IFM sarcomeres, including the assembly of thick filaments and sarcomere stability [18]. Furthermore, flightin has a significant impact on flying muscles, affecting myofilament stiffening and the muscle’s delayed stress response to stretching, as demonstrated by [19,20]. Flightin also plays a vital role in determining the length of thick filaments in IFM. An observation by [21] revealed that a *Drosophila* variant lacking flightin, known as fln0, exhibited IFM thick filaments over 40% longer than those found in the wild type. Additionally, the discovery of flightin’s capacity to bind to myosin rods has highlighted the critical nature of this interaction in the functioning of active muscles [22]. However, the molecular characterization and gene sequencing of flightin in most Vespidae species remain unclear.

The flightin gene has a size of 20 kDa and is associated with sarcomere and myosin function. The NH_2_-terminal region of flightin contains acidic amino acids, while the C-terminal region contains basic amino acids. Interestingly, the NH_2_-terminal 12-residue flightin sequence bears a resemblance to actin, indicating that flightin may regulate IFM contraction by influencing the interaction between myosin and actin [13]. The NH_2_-terminal site of flightin is considered to have low conservation but high-density phosphorylation sites. These phosphorylation sites have an impact on the stiffness, viscoelasticity, and power output, ultimately affecting insect flight [23,24]. In *Drosophila melanogaster*, flightin exists as a polyphosphorylated protein with 11 isoelectric variants, with 9 variants being generated through multiple phosphorylation events. Flightin is closely associated with thick myofilament structures, and the regulation of its phosphorylation/dephosphorylation is dependent on the presence of thin myofilaments. Studies have demonstrated that blocking the assembly of thick myofilaments inhibits the phosphorylation of flightin while blocking the assembly of thin myofilaments accelerates its phosphorylation [25]. The variation in phosphorylation sites, even among *Drosophila* and other arthropods, is of significant interest.

With limited availability of flightin gene sequencing data in winged insects in the NCBI database, we conducted the first phylogenetic analysis based on flightin gene sequencing in Hymenoptera. Currently, less than 80 species of arthropods have been sequenced for flightin. This research not only contributes to expanding the knowledge of flightin gene sequencing across various organisms but also provides valuable insights into evolutionary relationships. Furthermore, in this study, we performed functional characterization of the flightin protein, identified conserved sites, and determined its 3D structure. Interestingly, our findings revealed the absence of a transmembrane topology structure in the flightin protein. Lastly, we constructed a phylogenetic tree using our sequence data to enhance our understanding of the evolutionary relationships among different species.

## 2. Materials and Methods

### 2.1. Sample Collection

The samples were collected on 22 November 2023, in Qianyang County, Baoji City, Shaanxi, China, and then placed in a transparent beekeeping box in the laboratory. We used anatomical scissors for dissection. The desired part of the chest was placed separately in a 50 mL centrifugal tube, labeled, and placed in dry ice.

### 2.2. Reagents

Reagents were used to facilitate the experimental protocol. Noteworthy reagents were Trizol Extraction kit product no. B511321 (Sangon Biotech, Shanghai, China), M-MuL Vreverse Transcriptase, RNase H-, LA Taq manufactured by TaKaRa, product no. DRR02AG, Sangon Biotech (Shanghai, China) product no. B500517, Column DNA Gel Extraction Kit manufactured by Sangon Biotech (Shanghai, China), product no. B518131. Ethanol, chloroform, TAE buffer, DNA loading dye, dNTPs, and Primers were utilized for the amplification of the flightin gene.

### 2.3. Instruments

Instruments used in the experiments included PCR reaction amplifier manufactured by Canadian BBI, Stabilizer and current electrophoresis instrument manufactured by Shanghai Qite Analytical Instrument Co., Ltd. (Shanghai, China), Model (DYY-8), Centrifuge manufactured by Xiangyi Centrifuge Instrument Co., Ltd. (Changsha, China), Model (YXJ-2), Micro electrophoresis tank manufactured by Shanghai Jingyi Organic Glass Products Instrument Factory (Shanghai, China), Model (H6-1), Gel Imaging System manufactured by Gene Genius, UV–visible spectrophotometer manufactured by Hitachi (Tokyo, Japan), Model (U-3010), Clean Bench manufactured by Sujie Purification Equipment Factory (Suzhou, China), Model (SW-CJ-1D). Clustal Omega and MEGA_11.0.13 software, Water bath, Shaker/incubator, Bioinformatics hardware (computer, server), and Micropipettes Stereo/light microscope were used in the analysis.

### 2.4. RNA Extraction and cDNA Synthesis

RNA extraction was performed using the Trizol Extraction Kit. Subsequently, RNA detection was carried out by preparing a 1.5% agarose gel with 1× TAE electrophoresis buffer. The gel was visualized through UV transillumination and documented by photography to confirm RNA. 

Add the following reagents to a 0.2 mL PCR tube: 5 μL RNA sample, 1 μL random primers, 1 μL ddH_2_O. Heat the mixture at 70 °C for 5 min. Then, transfer the tube to an ice bath for 2 min. Centrifuge the tube and add the following reagents: 2.0 μL 5× *g* First-Strand Buffer, 0.5 μL 10 mmol dNTP, 0.25 μL RNase inhibitor, 0.25 μL Reverse Transcriptase, 10.0 μL total volume. Incubate the mixture at 42 °C for 60 min, followed by a 10 min incubation at 72 °C.

### 2.5. PCR Amplification of the Flightin Gene

Degenerate two primers were designed based on conserved regions of the flightin gene sequence from *V. velutina* (GenBank ID: LOC124952177). (P4) PCR primer was designed (sense 5′-ATAAAAGCAAGAAAGAAGGCAAGGT-3′ and anti-sense 5′ TATCGGC GGATCTCTTAATACTCTT-3′) for a product length of 1.2 kb bp. (P5) PCR primer was (sense 5′-CGCCCATTATTTCTCAATTATGACT-3′ and anti-sense 5′-TAAGTAAAAGTGCACGTGTATTTTATAAT-3′) designed for a product length of 500 bp. First, the P4 primer was used, then the P5 primer by the following process. PCR amplification of the *V. basalis* flightin gene fragments was accomplished in a 25 μL reaction volume holding a 10.1 μL template, using Taq DNA polymerase (Sangon Biotech, Shanghai, China). First, an initial denaturation phase was conducted for three minutes at 95 °C to start the thermal cycling process. A total of 33 cycles of denaturation (at 94 °C for 30 s), annealing (at 58 °C for 30 s), and extension (at 68 °C for 120 s) were then performed. The last extension phase was then carried out for seven minutes at 72 °C. The cycles were done multiple times. To visualize under UV light, the amplified PCR products were electrophoresed on a 1.7% agarose gel and treated with Green View nucleic acid dye [26,27]. The desired amplicon was excised and purified from the gel using a PCR purification kit (Sangon Biotech, China) for downstream applications.

### 2.6. Online ExPASy Tool

We used the online ExPASy tool for finding the molecular weight, isoelectric point, Conservative Domains, and Functional Sites. We opened ExPASy, searched for the ProtParam online analysis tool, opened the tool page, pasted the measured target amino acid sequence, and analyzed the amplified flightin. 

## 3. Results

### 3.1. PCR Result

Through PCR amplification and sequence splicing, a partial 1189 bp *V. basalis* flightin cDNA sequence was obtained. An analysis with the NCBI ORF Finder tool revealed the existence of an open reading frame with a base pair range of 160 to 609. It is anticipated that this open reading frame encodes a 150 amino acid polypeptide. In the agarose gel, the P4 primer band was approximately 1.2 kb bp in size, and the P5 band showed 500 bp (Figure 1). These results validate the PCR assay and confirm the successful formation of cDNA of the *V. basalis* flightin gene containing a full open reading frame for further characterization. The sequenced gene is provided in S4.

### 3.2. Molecular Weight Isoelectric Point Findings

The results show that the flightin gene encodes a peptide composed of 150 aa having a theoretically predicted protein (MW) of 18.05 kDa, and an isoelectric point (PI) of 5.84. There are 27 residues with a negative charge in total, made up of glutamic acid (Glu) and aspartic acid (Asp). On the other hand, arginine (Arg) and lysine (Lys) make up 24 of the residues with a positive charge. The percentage of amino acids in *V. basalis* flightin is shown in (Figure 2A).

### 3.3. Conservative Domains and Functional Sites

The results of the analysis for conserved domains on NCBI revealed that the amino acid sequences of *V. basalis* flightin do not possess any conserved sequence. Subsequently, the analysis of the functional sites of the flightin protein in the wasp showed that the flightin contained one casein kinase II phosphorylation site, four N-myristoylation sites, five protein kinases C phosphorylation sites, and single tyrosine kinases phosphorylation site 2, with four different patterns, as shown in Table 1.

### 3.4. Prediction of Transmembrane Topology

The online analysis tool TMHM Server v2.0 was used to predict the transmembrane topological structure of the flightin protein of *V. basalis*. The research findings showed that the *V. basalis* flightin has no transmembrane topological structure, as shown in Figure 2D.

### 3.5. Prediction of Secondary Structures

PBIL’s online analysis tool SOPMA was used to predict the secondary structure of flightin protein sequences. The research findings showed that the flightin protein sequence α-helix (Hh) accounts for approximately 41.33% of an extended strand (Ee) accounts for approximately 6.67%, β-turn (Tt) accounts for approximately 3.33%, while the random coil (Cc) structure accounts for approximately 48.67% (Figure 2E).

### 3.6. 3D Structure Prediction

SWISS-MODEL online analysis tool in ExPASy was searched and a 3D structure model of the flightin protein was constructed. The 3D structure of flightin was built based on the template of the bumblebee Bombus model (Template: 8ew5.1. A), achieving a sequence identity of 75.68% and a sequence similarity of 0.55; GMQE value: 0.52, QMEANDis Co Global: 0.57 ± 0.08 (Figure 2B). The rationality of the 3D structural model of the simulated flightin was checked using the Ramachandran diagram shown in (Figure 2C). The results showed that the dihedral angle and three-dimensional concept of the flightin three-dimensional structure model are reasonable and consistent with two dihedral angles, ϕ and ψ. The basic requirements for distribution (Figure 2B) indicated that the constructed 3D structure was reasonable and reliable.

### 3.7. Protein Homology Analysis

Making use of the predicted amino acid sequence found in the wasp’s flightin protein, a BLAST search was performed on NCBI. As a result, amino acid sequences of the flightin protein of insects were screened out; the detailed accession numbers are provided in (Appendix A), including 8 species of Vespidae, 17 species of Apoidae, and 13 species of Formicidae. The sequence of amino acids in the flightin protein of *V. basalis* was compared to the sequences obtained from NCBI, and the results showed that *V. basalis* had the highest sequence identity with the Vespinae of the Vespidae, with up to 99.33% with the *V. mandarinia*, 98% with the V. crabro, 96.97% with the V. velutina, 90% with the Vespula vulgaris, 89.33% with the V. pensylvanica. The sequence identity was 70% with the Polistes dominula and P. fuscatus of the Polistinae, 70–80% with the Apoidea, and 70–80% with the Formicidae.

The results through the T-COFFFEE webpage showed that 29 completely conserved regions had 67 conserved sites and 30 conserved replacement sites. The conserved amino acid sites were sequentially MWWPLPPYNHHWVRPLFLNYYYRNYYDIYLRGR EPQEWAERRYDNSKDR YSYHTRAYYSKYQI.

We used MEGA_11.0.13 software to construct a phylogenetic tree by ML (Maximum Likelihood Estimate) method and performed 1000 bootstraps evaluations to show branch support. Among the species with known flightin gene sequences, *V. basalis* has a high homology with species of the same family, among which *V. basalis* has the highest homology and closest relationship with *V. mandarinia* (Figure 3). The protein homology is also shown in Appendix A.

## 4. Discussion

In this study, we obtained the partial cDNA sequence of the flightin gene in *V. basalis* using PCR technology. The sequence had a total length of 1189 bp, and the open reading frame was predicted to be 453 bp, encoding a protein of 150 amino acids. The calculated molecular weight of the flightin protein was 18.05 kDa, with an isoelectric point of 5.84. Previous research on *Drosophila melanogaster* identified flightin as a polyphosphorylated myofibrillar protein with a size of 20 kDa [14]. In *V. basalis* flightin, we observed one phosphorylation site for casein kinase II, four N-myristoylation sites, five sites of phosphorylation for protein kinase C, and one site of phosphorylation for tyrosine kinase 2, with four different modes. *Drosophila* flightin protein contains consensus phosphorylation sites for casein kinase II, cAMP and cGMP-dependent protein kinases, and protein kinase C in its basic sequence [13]. Studies by Lemas et al. demonstrated dynamic changes in the phosphorylation state of flightin during a flight in *D. melanogaster*, while some phosphorylation sites remained invariant. These findings provide insights for further analysis of flight regulation and function [28]. Although *V. basalis* and *Drosophila* share phosphorylation sites for casein kinase II and protein kinase C, the number and types of other phosphorylation sites differ. Similar studies have suggested that the location and number of flightin phosphorylation sites may be involved in its function and regulation in muscle fibers.

The analysis of flightin’s secondary structure prediction in *V. basalis* revealed a predominant presence of α-helix and random coil structures. This observation was supported by the 3D structure model, which indicated a higher abundance of α-helix elements. These findings are consistent with the secondary structure of flightin in *Drosophila*, which exhibits two helical regions [13]. Cryo-electron microscopy studies by Menard et al. on the thick myofilament of Lethocerus identified a specific red density in the contact area of the thick myofilament skeleton, suggesting the presence of flightin [13,29,30]. Our findings suggest that flightin facilitates the contact between multiple myosin dimers, promoting their orderly assembly and stabilizing myosin coils. This mechanism enhances the levels of individual myosin dimers, thereby regulating muscular contraction. These results emphasize the critical role of flightin in the regulation of insect flight muscles.

Comparing the flightin amino acid sequences of *V. basalis* with other species, we found 67 conserved sites, 29 completely conserved regions, and 30 conserved replacement sites. In comparison, the flightin proteins of the brown planthopper [31] and *Spodoptera litura* [32] have 14 and 31 completely conserved regions, respectively. The flightin protein sequence shows a relatively conservative nature, with the critical factors involved in muscle regulation primarily located in the middle region. Phylogenetic analysis revealed a close relationship between *V. basalis* and *V. mandarinia*, both belonging to the same genus, while *V. crabro* Linnaeus exhibited the lowest homology and most distant kinship. Larger-bodied species like *V. magnifica, V. ducalis*, and *V. mandarinia* formed a separate clade in the traditional phylogenetic tree. *V. mandarinia* showed the closest relationship to *V. magnifica* but was quite distant from *V. basalis* [33,34]. Despite the similarity in flightin phosphorylation sites, it is important to note that flightin similarity alone may not directly account for the observed similarities in pectoral muscles and flying ability. Further investigations are required to fully understand the specific mechanisms underlying these similarities.

Flightin’s key role in insect flight and its multiple functions have been explored in previous studies. Chakravorty et al. used transgenic technology to create fln∆N62 *Drosophila* flies lacking the N-terminal 62 amino acids of the flightin protein. They analyzed the changes in muscle structure and insect flight ability and found that the N-terminal domain is essential for adjusting the insect muscle lattice structure and flight mechanics. Deletion of this domain resulted in abnormal muscle structure and reduced flight ability and affected the expression of courtship signals in *Drosophila*. This finding has important implications for biomechanical and biophysical research [8]. RNA interference (RNAi) experiments targeting the flightin gene in Nilaparvata lugens showed that downregulating flightin expression significantly reduced flight ability, and, in some cases, eliminated flight ability in the insects [30]. Chen et al. successfully disrupted the standard sarcomere structure of the dorsal longitudinal muscle in male adult rice leaf rollers through RNA interference experiments targeting the flightin gene. This disruption prevented males from producing routine courtship calls, and, consequently, they were unable to attract females for mating [9]. After gene interference and related research verification, it has been found that flightin is a key component in flight insects. This study analyzed the molecular characterization flightin in *V. basalis.* It is necessary to conduct RNA interference experiments and verify the related mechanisms. Therefore, additional investigations are required to further validate the important function of flightin in wasp flight.

## 5. Conclusions

In conclusion, the partial cDNA sequence analysis of the flightin gene in *V. basalis* provides valuable insights into the structure and function of flightin in insect flight muscles. The identified phosphorylation sites and conserved regions shed light on the regulation and evolutionary conservation of flightin across different insect species. The predominant presence of α-helix structures and the proposed role of flightin in promoting myosin assembly and stabilizing myosin coils suggest its crucial involvement in muscle contraction and flight mechanics. Further studies, including functional analyses and comparative studies across a wider range of insect species, are necessary to fully understand the complex mechanisms underlying flightin’s role in insect flight.

## Figures and Tables

**Figure 1 animals-14-00978-f001:**
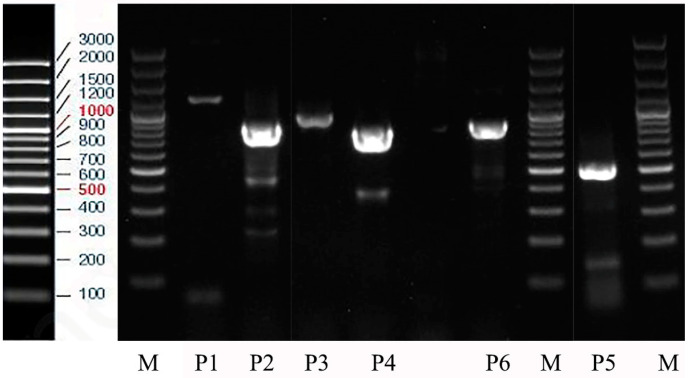
1.7% Agarose gel electrophoresis analysis product of flightin gene. P4 is an open reading frame showing a band at approximately 1.2 bp kb, and P5 shows the flightin band at approximately 500 bp, M = 100–3000 bp ladder-K. P1, P2, P3 primer details are given in Appendix A.

**Figure 2 animals-14-00978-f002:**
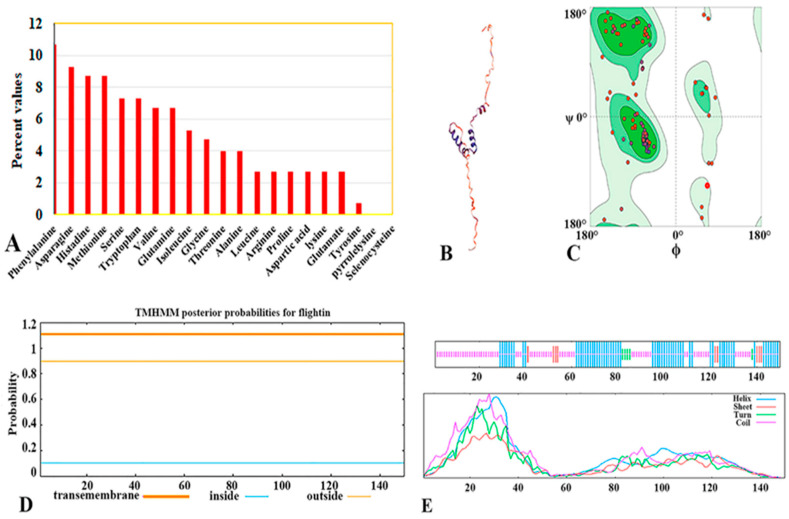
(**A**) The percentage values of amino acids in the flightin of *V. basalis*. (**B**) 3D structure of the flightin of the *V. basalis* and (**C**) Ramachandran. (**D**) Prediction of the transmembrane topological structure of flightin protein. (**E**) SOPMA-2D structure prediction. (Blue shows alpha helix, red shows extended strand, green shows beta-turn, and yellow shows random coil).

**Figure 3 animals-14-00978-f003:**
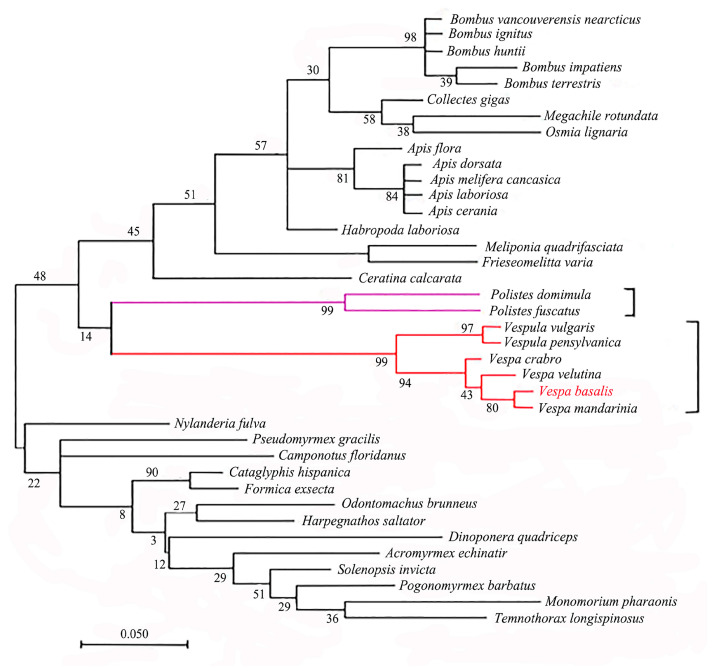
Phylogenetic tree based on flightin gene cDNA sequence. Branch numbers indicate bootstrap support percentages (1000 replicates). The scale bar represents 0.050 substitutions per site, reflecting the average genetic divergence per branch.

**Table 1 animals-14-00978-t001:** Functional site analysis of flightin protein sequence.

Functional Site	Flightin	
	Site	Sequence Signature
Casein kinase II phosphorylation site	14–17	TapE
N-myristoylation site	24–29	GAkeGG
	28–33	GGaeGA
	29–34	GAegAA
	32–37	GAapGE
Protein kinase C phosphorylation site	102–104	TmR
	114–116	StK
	115–117	TkR
	130–132	TpR
	143–145	SlK
Tyrosine kinase phosphorylation site 2	116–123	KrsaDmkY
	4 different patterns found	

## Data Availability

Data are contained within the article and Appendix A.

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
