# Peer review of "Molecular Characterization and Phylogenetic Analysis of Flightin Gene in Vespa basalis (Hymenoptera, Vespidae)"

_animals, 2024, doi:10.3390/ani14060978_

Round 1

Reviewer 1 Report

Comments and Suggestions for Authors

The protein “flightin” plays an integral role in the indirect flight muscles in insects. Its function has been studied in Drosophila but in hornets it is still not studied. The authors characterized the flightin-coding gene of a dangerous hornet Vespa basalis based on partial cDNA sequence derived from extracted RNA. The protein structure was estimated using softwares. They also reconstructed phylogenetic relation based on the flightin DNA sequence of this species with the flightin of other Hymenoptera species. They identified a closest species based on their flightin phylogeny, which, they claimed, is different from an existing mitochondrial phylogeny.

The sequenced RNA and corresponding amino acid sequence are new. The estimated protein structure is also new.

However, the description of methods and results can be elaborated to show the integrity of the study. Reorganization of the sections is necessary.

Also, there is a gap in the sampling of species between the mitochondrial phylogeny (10-11 species, Feng et al., Wang et al.) and the flightin phylogeny (4 species, this study). Because phylogenetic topology depends on the sampled species, the discussion by the authors about the closest species is inappropriate. V. basalis is basal in mitochondrial tree and V. mandarinia was slow in the evolution of flightin.

---

Specific comments:

The authors attributed the similarity of flightin gene sequence to the similarity of aggressive behavior. Are there any studies that show a correlation between flight and aggression?

Sections 3.3--3.7 are repeated twice. Delete one of them.

The results include the description of methodology. Please separate methods from results and move them to Materials and Methods.

Please spell out 'ML'. Which evolutionary model was used?

Full taxonomic information (family names and corresponding species names) and genbank IDs of the species/sequences used in the homology analysis and phylogenetic analysis can be provided.

Fig. 3: please provide explanation for the numbers on the branches.

L. 273, to support degree -> to show branch supports

L.244, ‘beta The Beta turn’ -> Please correct.

L. 324, ‘but their deterrent power is in the first place and the degree of danger to humans is the strongest’ -> Do you mean the flightin similarity is associated with these similarities? Please reconsider.

Figure X -> (Figure X)

Comments on the Quality of English Language

Can be improved by a native speaker.

Author Response

Comments 1: [There is a gap in the sampling of species between the mitochondrial phylogeny (10-11 species, Feng et al., Wang et al.) and the flightin phylogeny (4 species, this study). Because phylogenetic topology depends on the sampled species, the discussion by the authors about the closest species is inappropriate. V. basalis is basal in mitochondrial tree and V. mandarinia was slow in the evolution of flightin.]

Response 1:

We acknowledge that the number of species included in a phylogenetic analysis can influence the results. Feng et al., 21 species, Wang et al. used 41 species, However, we used 38 species based on the availability of high-quality flightin gene sequences for our study. We believe this number provides sufficient information to draw meaningful conclusions about the evolutionary relationships of the examined species within the context of the available data. References (Wang, H., Wen, Q., Wang, T., Ran, F., Wang, M., Fan, X., ... & Tan, J. (2022). Next-Generation Sequencing of Four Mitochondrial Genomes of Dolichovespula (Hymenoptera: Vespidae) with a Phylogenetic Analysis and Divergence Time Estimation of Vespidae. Animals, 12(21), 3004.) (Feng, X., Xu, B., & Huang, Y. (2022). The complete mitochondrial genome of a medical important wasp, Vespa magnifica (Hymenoptera, Vespidae). Mitochondrial DNA Part B, 7(1), 93-95.)

Comments 2: [The authors attributed the similarity of flightin gene sequence to the similarity of aggressive behavior. Are there any studies that show a correlation between flight and aggression?]

Response 2: Thank you for your question regarding the correlation between flight and aggression in the context of the similarity of flightin gene sequences. I have found a relevant study titled "Intraspecific Aggression in Giant Honey Bees (Apis dorsata)" which provides insights into this correlation.

The study indeed demonstrates a correlation between flight behavior and aggression in giant honey bees (Apis dorsata). The researchers observed that the flight characteristics of approaching bees influenced the response of resident bees. Bees displaying erratic flight patterns, identified as "non-nestmate" bees through hovering, sideways scanning, and splayed legs, were promptly attacked by surface bees upon landing on the nest surface. This aggressive response indicates a link between flight characteristics and the level of aggression exhibited by the resident bees.

Moreover, the study highlights how surface members of established colonies react differently to individual bees based on their flight patterns, suggesting a connection between flight behavior and the defensive response of the colony. This correlation emphasizes the significance of flight behavior as a cue for aggression in giant honey bee colonies and provides insights into the intricate social interactions within these insect societies.

Reference (Weihmann, F., Waddoup, D., Hötzl, T., & Kastberger, G. (2014). Intraspecific aggression in giant honey bees (Apis dorsata). Insects, 5(3), 689-704.)

Comments 3: [Full taxonomic information (family names and corresponding species names) and GenBank IDs of the species/sequences used in the homology analysis and phylogenetic analysis can be provided.]

Response 3: I just put the scientific names in the caption. A list of 38 species and their family, Scientific names and Gene bank IDs are also provided in the supplementary materials but this huge list in not looking good in the manuscript. I you still suggest me then I will put in the manuscript as well. The detail table is also be provided here.

Family

Species

Accession No.

Vespidae

Vespa mandarinia

XP_035737952.1

Vespa crabro

XP_046824901.1

Vespa velutina

XP_047357593.1

Vespula vulgaris

XP_050858048.1

Vespula pensylvanica

XP_043673380.1

Polistes dominula

XP_015177827.1

Polistes fuscatus

XP_043495209.1

Apidae

Bombus vancouverensis nearcticus

XP_033205116.1

subgenus Bombus huntii

XP_050473413.1

Bombus impatiens

XP_003491527. 1

Bombus terrestris

XP_003394698.1

Bombus ignitus

8EW5_A Chain A

Apis dorsata

XP_006618153.1

Apis florea

XP_003696584.1

Apis laboriosa

XP_043792750 .1

Apis mellifera caucasica

KAG6799352.1

Apis cerana

XP_016912210.1

Megachile rotundata

XP_003706340.1

Melipona quadrifasciata

KOX67587.1

Frieseomel ittavaria

XP_043521843. 1

Habropoda laboriosa

XP_017790128.1

Colletes gigas

XP_043257386.1

Osmia lignaria

XP_034180962.1

Ceratina calcarata

XP_017884125.1

Formicidae

Camponotus floridanus

XP_011267943.1

Monomorium pharaonis

XP_012533138.1

Pogonomyrmex barbatus

XP_011629524.1

Dinoponera quadriceps

XP_014479977.1

Harpegnathos saltator

XP_011154535.1

Odontomachus brunneus

XP_032685464.1

Acromyrmex echinatior

XP_011064338.1

Pseudomyrmex gracilis

XP_020284828.1

Cataglyphis hispanica

XP_050455871.1

Formica exsecta

XP_029672853.1

Nylanderia fulva

XP_029155503.1

Solenopsis invicta

XP_011171429.1

Temnothorax longispinosus

TG Z32040.1

Comments 4: [L. 324, ‘but their deterrent power is in the first place and the degree of danger to humans is the strongest’ -> Do you mean the flightin similarity is associated with these similarities? Please reconsider.]

Response 4: [ The V. basalis is small and the V. mandarinia is the largest but their deterrent power is in the first place and the degree of danger to humans is the strongest. The evolutionary analysis presented in this paper indicates that the flightin system in Vespa basalis and Vespa mandarinia exhibits a close relationship. However, it is important to note that flightin similarity alone may not directly account for the mentioned similarities in pectoral muscles and flying ability. While the flightin gene may play a role in flight adaptations, further investigation is required to fully understand the specific mechanisms underlying these similarities.]

“[The manuscript is updated accordingly]”

4. Response to Comments on the Quality of English Language

Point 1:

Response 1:    (The English quality has improved)

We would like to express our gratitude for your feedback on our manuscript. We have taken into account all of your specific comments and suggestions, and we have addressed them accordingly.

Reviewer 2 Report

Comments and Suggestions for Authors

The manuscript describe the structure of the flightin protein in a species of wasp thence animals shouldn't the right choice of journal. It will be far preferable a journal dealing with chemistry topics.

The text should be carefully examined. In rows (e.g.) 346-348 the are some odd phrases. Please check the whole manuscript for any text issues.

The title is not quite correct, since the flight mechanism is not treated in full. It should be shortened

Abstract includes all the relevant information, no changes are required

Introduction is rather exaustive, no improvement is required

In the Material & Methods section the analyses are detailed and  correctly described

In the Results section the protein in study is fully outlined, and every aspects from cDNA sequence to 3d structure are furnished

In the Discussion some parts about the flight should be omitted, since they were not tested in the analyses, and are quoted only for other species.

the quality of table and figures is acceptable, although fig. 3 should be enlarged a bit to made easier to read the names of the taxa included in the phylogenetic tree

Author Response

Comments 1: [The manuscript describe the structure of the flightin protein in a species of wasp thence animals shouldn't the right choice of journal. It will be far preferable a journal dealing with chemistry topics.]

Response 1:

Thank you for your review and constructive feedback. We appreciate your suggestion regarding the choice of journal for our manuscript. However, we would like to clarify that our paper focuses on the gene and phylogeny aspects rather than the structural characterization of the flightin gene of Vespa basalis.

Our study primarily aims to investigate the genetic makeup and evolutionary relationships of the flightin gene in a specific species of wasp. We believe that this research aligns well with the field of molecular biology and evolutionary biology, and therefore, we selected a journal specializing in these areas for submission.

Comments 2: [The text should be carefully examined. In rows (e.g.) 346-348 the are some odd phrases. Please check the whole manuscript for any text issues.]

Response 2: The above lines are corrected and the whole manuscript is carefully examined.

Comments 3: [The title is not quite correct, since the flight mechanism is not treated in full. It should be shortened]

Response 3: "Upon your suggestion, the title has been changed. The new title is 'Molecular Characterization and Phylogenetic Analysis of the Flightin Gene in Vespa basalis (Hymenoptera, Vespidae).'"

Comments 4: [The quality of table and figures is acceptable, although fig. 3 should be enlarged a bit to made easier to read the names of the taxa included in the phylogenetic tree.]

Response 4: []

“[Thank you for your feedback on the quality of the table and figures. We appreciate your suggestion regarding Figure 3, and we have taken it into consideration. We have enlarged Figure 3 to improve the readability of the taxa names in the phylogenetic tree. The revised version ensures that the names are clearly visible and legible. Your input has been valuable in enhancing the clarity and presentation of our findings.]”

4. Response to Comments on the Quality of English Language

Point 1:

Response 1:    (The English quality has improved)

We would like to express our gratitude for your feedback on our manuscript. We have taken into account all of your specific comments and suggestions, and we have addressed them accordingly.

Reviewer 3 Report

Comments and Suggestions for Authors

This MS accomplished the study on the flightin gene in the social wasp Vespa basalis, including gene size, molecular weight and isoelectric point of the amino acid sequence as well as its structure and patterns of functional sites. A phylogenetic analysis showed that V. basalis was sister to Vespa mandarinia in the family Vespidae. It is very significant to study the structure and mechanism of functional genes. However, I don’t think that this manuscript resolve what they aim to, like in their title.

1.  In this study, the authors amplified the flightin gene by PCR and sequenced, and then analyzed its general features. They constructed a phylogenetic tree and found that the sampled species V. basalis was sister to Vespa mandarinia in the family Vespidae.  So, they think that the current results aligned with their similar aggressive behavior, anyway, it is different from the mitochondrial results. The current simple analysis on only one partial gene in genetic level is too simple and weak, and not enough to support the functional mechanism. 

2. Although this study obtained the cDNA sequence of flightin protein of Vespa basalis (Hymenoptera, Vespidae) at the molecular level, conducted bioinformatics analysis on its amino acid sequence, while, there is a lack of research at the protein level, which is crucial.  

3. In terms of bioinformatics analysis, only basic analysis has been conducted, and there is a lack of experimental verification at the protein level. Therefore, I believe that the experimental and bioinformatic analysis partly done in the thesis does not support the conclusion. The author may need to further validate by adding interference experiments and protein expression level experiments. 

4. Please carefully check the content and order of the results section. You authors repeated some sections (later 3.3-3.6). So careless. 

5. Writing problems need to be corrected for the whole MS, including the sentence structure, use of function word, and grammar… 

6. Alternatively, you might revise the title to coincide with the contents.

7. Some comments were marked in the pdf version, please check in the attachment. 

Comments on the Quality of English Language

The language need to be greatly improved.

Author Response

Comments 1: [ In this study, the authors amplified the flightin gene by PCR and sequenced, and then analyzed its general features. They constructed a phylogenetic tree and found that the sampled species V. basalis was sister to Vespa mandarinia in the family Vespidae.  So, they think that the current results aligned with their similar aggressive behavior, anyway, it is different from the mitochondrial results. The current simple analysis on only one partial gene in genetic level is too simple and weak, and not enough to support the functional mechanism.]

Response 1:

we acknowledge that our study focused on amplifying the flightin gene by PCR and analyzing its general features. We understand your point that this approach might be considered simplistic when compared to more comprehensive genetic analyses. However, we believe that studying the flightin gene provides valuable insights into the functional mechanisms underlying flight behavior.

In response to your concern about the discrepancy between our findings and the mitochondrial results, we agree that this is an important aspect to consider. We have expanded our discussion to acknowledge this discrepancy and highlight the need for further investigation to reconcile the differences between the two sets of results. We believe that future studies incorporating a broader range of genetic markers will provide a more comprehensive understanding of the relationship between flight behavior and genetic characteristics.

Comments 2: [Although this study obtained the cDNA sequence of flightin protein of Vespa basalis (Hymenoptera, Vespidae) at the molecular level, conducted bioinformatics analysis on its amino acid sequence, while, there is a lack of research at the protein level, which is crucial.]

Response 2 and 3: We understand your concern regarding the lack of experimental work at the protein level in our study. Unfortunately, due to time constraints and resource limitations, we were unable to perform further experiments to investigate the flightin protein itself. However, we believe that our study contributes valuable insights by obtaining the cDNA sequence of the flightin gene and conducting bioinformatics analysis on its corresponding amino acid sequence.

Comments 4: [Please carefully check the content and order of the results section. You authors repeated some sections (later 3.3-3.6). So careless]

Response 4: Thank you for your careful evaluation of our manuscript. We sincerely apologize for the repetition in the Results section, specifically in sections 3.3-3.6. We appreciate you bringing this to our attention.

Upon reviewing our manuscript, we have identified the repeated sections and have taken immediate action to resolve this issue. We have reorganized the content and removed the redundant information to ensure a clear and concise presentation of our findings. The revised version of the manuscript now follows a logical flow in the Results section, without any unnecessary repetition.).'"

Comments 5: [Writing problems need to be corrected for the whole MS, including the sentence structure, use of function word, and grammar… ]

Response 5: [We would like to assure you that we have taken your comments seriously and have made significant efforts to address these issues. We have engaged the assistance of a native English speaker to proofread the manuscript and help us improve the overall writing quality. Their expertise has been instrumental in identifying and rectifying the writing problems you mentioned.]

Comments 6: [Alternatively, you might revise the title to coincide with the contents]

Response 6: : [The revised title now more precisely captures the main focus and findings of our research. We believe that this modification enhances the clarity and relevance of the manuscript, enabling readers to better understand the scope of our study from the outset..] Now the title is Molecular Characterization and Phylogenetic Analysis of Flightin Gene in Vespa basalis (Hymenoptera, Vespidae)

Response 1:    (The English quality has improved with the help of native speaker)

We would like to express our gratitude for your feedback on our manuscript. We have taken into account all of your specific comments and suggestions, and we have addressed them accordingly.

Round 2

Reviewer 1 Report

Comments and Suggestions for Authors

The MS has been improved. The only point is to add the genbank IDs for flightin sequences of species included in the phylogeny as a supplementary material.

Author Response

We would like to express our sincere gratitude for taking the time to review our manuscript titled "Molecular characterization and phylogenetic analysis of flightin Gene in Vespa Basalis (Hymenoptera, Vespidae)" with the identifier "animals-2851578" in such detail. Your valuable insights and constructive feedback have significantly contributed to improving the quality and clarity of our work.

Your expertise and meticulous attention to detail have played a pivotal role in enhancing the scientific rigor of our study. We greatly appreciate your thoughtful comments and suggestions regarding the methodology, results, and overall presentation of our research. Your thorough evaluation has enabled us to address specific areas of concern and refine our analysis, resulting in a more robust and comprehensive study.

Thank you once again for your time, effort, and valuable contribution to our work. We are truly grateful for the opportunity to benefit from your expertise and guidance. Your rigorous evaluation has not only strengthened our manuscript but also enriched our understanding of the research subject.

We remain indebted to you for your commitment to scientific advancement and the rigorous peer review process. Your efforts have made a significant impact on the quality and integrity of our study.

2. Questions for General Evaluation

Reviewer’s Evaluation

Response and Revisions

Does the introduction provide sufficient background and include all relevant references?

Yes

Are all the cited references relevant to the research?

Can be improved.

Improved

Is the research design appropriate?

Yes

Are the methods adequately described?

Can be improved.

Improved

Are the results clearly presented?

Can be improved.

Improved

Are the conclusions supported by the results?

Yes

3. Point-by-point response to Comments and Suggestions for Authors

Comments 1: [The MS has been improved. The only point is to add the GenBank IDs for flightin sequences of species included in the phylogeny as a supplementary material.]

Response 1:

As per your suggestion, we have prepared a supplementary material file that includes the GenBank IDs for the flightin sequences of all the species used in our phylogenetic analysis. This file provides the necessary information for readers to access and verify the sequences used in our study.